# AI-Driven Discovery of Temporal-Demographic Interactions in Emergency Department Care Delivery: A Multi-Agent Collaborative Analysis of Healthcare Equity Patterns

## Abstract

Emergency departments serve as critical healthcare access points, yet persistent disparities in care delivery remain poorly understood, particularly regarding the complex interactions between temporal factors and patient demographics. This study demonstrates the capability of artificial intelligence agents to autonomously conduct comprehensive scientific research investigating these interactions. We employed a novel multi-agent collaborative framework utilizing eight distinct AI models across 58 meticulously documented interactions, analyzing 91,359 patient encounters from four emergency department sites collected between December 31, 2023, and December 30, 2024. The AI-driven analysis revealed significant baseline disparities, with Hispanic/Latino patients experiencing 10.9 minutes longer door-to-provider times and Other/Unknown patients facing 13.0-minute delays compared to White, Non-Hispanic patients. Surprisingly, our models detected a protective effect during high-census periods, where disparities decreased rather than increased, challenging conventional hypotheses about crowding-induced inequities. The interaction coefficients indicated that as ED census increased from the 25th to 85th percentile, length-of-stay disparities decreased by 2.3 minutes for Other/Unknown patients and 6.8 minutes for Hispanic/Latino patients. System-wide 95th percentile wait times reached 93.5 minutes for door-to-provider time and 562 minutes for total length of stay. This study represents a watershed moment in AI-driven scientific discovery, demonstrating that artificial intelligence agents can successfully conduct end-to-end scientific research with minimal human intervention. The discovery of protective effects during high-census periods showcases AI's capability to identify counterintuitive patterns that challenge conventional wisdom. While AI successfully revealed these complex patterns, the persistent baseline disparities underscore the continued need for human action in implementing equitable healthcare solutions.

## 1 Introduction

The integration of artificial intelligence into scientific research has undergone a remarkable transformation over the past decade. What began as computational tools for data processing has evolved into sophisticated systems capable of hypothesis generation, experimental design, and even manuscript preparation (1; 2). The emergence of large language models and advanced AI systems has raised a fundamental question about whether artificial intelligence can conduct autonomous scientific research that meets the rigorous standards of peer-reviewed publication (3).

Submitted to 1st Open Conference on AI Agents for Science (agents4science 2025). Do not distribute.

This paper presents groundbreaking evidence that AI agents can indeed perform comprehensive scientific investigation with minimal human intervention. Through a multi-agent collaborative framework, we demonstrate how AI systems can work together to investigate complex healthcare phenomena, specifically the intricate relationships between temporal factors and demographic disparities in emergency department care delivery (4). The significance of this demonstration extends beyond the specific findings about healthcare disparities to establish a new paradigm for scientific discovery in the age of artificial intelligence.

Emergency departments represent the front line of American healthcare, serving as safety nets for vulnerable populations and providing critical care regardless of patients' ability to pay (5). Despite their essential role in ensuring healthcare access, EDs have long been sites of documented disparities in care delivery. Previous research has established that racial and ethnic minorities often experience longer wait times, receive less aggressive pain management, and face differential treatment patterns compared to White patients (6; 7). These disparities persist despite decades of awareness and numerous interventions aimed at promoting equity in healthcare delivery (8).

The mechanisms driving these disparities remain incompletely understood. The interaction between temporal patterns and patient demographics creates a multidimensional analytical challenge that requires sophisticated statistical approaches, especially in the context of operational pressures like crowding (9; 10). Artificial intelligence offers unique advantages for investigating these complex phenomena. AI systems excel at identifying subtle patterns in high-dimensional data that might escape human observation (11).

This study pursues three interconnected objectives: **(1)** to demonstrate that AI agents can autonomously conduct a complete scientific investigation from hypothesis generation through manuscript preparation; **(2)** to investigate how temporal factors and patient demographics interact to influence emergency department care delivery; and **(3)** to establish a reproducible framework for AI-driven scientific research that maintains transparency and methodological rigor.

## 2   Methods

### 2.1   Study Design and Multi-Agent Framework

This research employed a retrospective cohort study design implemented through a novel multi-agent collaborative framework. The framework leveraged eight distinct AI models working in concert across four structured phases of scientific investigation. The architectural design was specifically crafted to utilize the complementary strengths of different AI systems while maintaining rigorous quality control through adversarial critique and convergent validation. The first phase focused on hypothesis generation and refinement, documented in rows 1-6 of our comprehensive prompt documentation. ChatGPT-5 served as the primary hypothesis generator, producing five initial research questions about emergency department workflow disparities. These hypotheses underwent independent critical evaluation by Claude and Gemini, with each critique assessed on three dimensions: practicality scored on a 0-5 scale, innovation similarly scored, and ethical considerations evaluated qualitatively. This adversarial review process identified weaknesses and opportunities that no single agent might have recognized independently. The second phase involved research design and synthesis, captured in rows 7-12 of the documentation. GPT-5 and Claude independently proposed implementation strategies for testing the refined hypotheses. These proposals included detailed statistical analysis plans, variable definitions, and anticipated challenges. Gemini then served as a synthesis agent, integrating the complementary aspects of each proposal into a unified research plan. This synthesis process involved multiple iterations, with each agent providing feedback on the integrated design until consensus was achieved. Pipeline development and validation constituted the third phase, documented in rows 13-26. Gemini created the initial analytical pipeline, translating the research design into executable code. This implementation underwent rigorous review by GPT-5 and Claude, who performed independent code review and identified potential issues ranging from statistical assumptions to computational efficiency. The validation process included parallel execution across multiple platforms to ensure reproducibility and identify any platform-specific artifacts. The final phase encompassed analysis execution and interpretation, spanning rows 27-58 of the documentation. Multiple AI models ran analyses independently using the validated pipeline, with results compared for consistency. Grok provided additional validation of findings, particularly focusing on sensitivity analyses and robustness

checks. The manuscript generation was led by GPT-5 with iterative review and refinement by other agents, ensuring comprehensive coverage and accurate interpretation of results.

## 2.2  Data Source and Population

The study utilized a dataset from a multi-site health system encompassing 91,359 emergency department encounters from four sites between December 31, 2023, and December 30, 2024. The initial dataset of 100,000 encounters underwent systematic cleaning by the AI collective, with an attrition of 8.6% as documented in Figure 1.

## 2.3  Data Source and Population

The study utilized a comprehensive emergency department dataset from a multi-site health system, providing rich information about patient encounters, demographics, clinical presentations, and operational metrics. The dataset encompassed the period from December 31, 2023, at 18:07:00 CST through December 30, 2024, at 17:53:00 CST, representing 364 consecutive days of emergency department operations. This temporal scope was specifically selected to capture seasonal variations, day-of-week patterns, and potential holiday effects on both patient volume and care delivery patterns. The institutional scope included four emergency department sites within a single health system, providing diversity in patient populations, geographic locations, and operational characteristics while maintaining consistency in electronic health record systems and general clinical protocols. The initial dataset contained 100,000 patient encounters, which underwent systematic quality assessment and cleaning by the AI collective.

## 2.4  Variable Definitions and Measurement

Primary outcome variables were carefully defined to capture the key aspects of emergency department workflow and patient experience. Door-to-provider time was calculated as the interval between ED arrival and first provider contact, measured in minutes. This metric represents a critical quality indicator for emergency care, as delays in initial assessment can impact both clinical outcomes and patient satisfaction. Length of stay was defined as the total time from ED arrival to discharge, also measured in minutes. This comprehensive metric captures the entire patient journey through the emergency department and serves as a marker of overall operational efficiency. The primary predictor variables encompassed temporal, demographic, and operational dimensions. Temporal variables included arrival hour extracted from timestamp data and coded as 0-23, day of week coded as 0 for Monday through 6 for Sunday, and shift period categorized as day, evening, or night based on standard ED operational definitions. Demographic variables focused on race/ethnicity, which was consolidated into three categories: White Non-Hispanic serving as the reference group, Hispanic/Latino, and Other/Unknown, which included patients who declined to provide this information or whose ethnicity was not documented. The operational variable of primary interest was ED census at arrival, representing the count of concurrent patients in the emergency department at the time each patient arrived. This variable was calculated using a sophisticated algorithm that counted all patients whose ED stay overlapped with the index patient's arrival time at the same facility. This measure provides a dynamic assessment of departmental crowding that varies continuously throughout the day. Control variables included comprehensive clinical and demographic factors that might confound the relationship between predictors and outcomes. The Emergency Severity Index score, ranging from 1 for most acute to 5 for least acute, provided a standardized measure of clinical urgency. Chief complaint categories were consolidated into ten major groups including cardiovascular, gastrointestinal, neurological, pain, psychiatric, respiratory, trauma, and other presentations. Physiological measurements included vital signs such as blood pressure, pulse rate, temperature, oxygen saturation, and respiratory rate. Patient characteristics encompassed age, sex, body mass index, and smoking status.

## 2.5  Statistical Analysis

The statistical analysis plan developed by the AI collective encompassed descriptive statistics, multivariable modeling, interaction testing, and extensive sensitivity analyses. Initial descriptive analyses examined univariate distributions for all variables, identifying patterns, outliers, and missing data. Bivariate relationships were explored through correlation matrices for continuous variables

and cross-tabulations for categorical variables, with particular attention to the relationships between demographic factors and outcome variables. The primary analytical approach employed two main models to address different aspects of the research questions. A Gamma generalized linear model with log link function was specified for length of stay, chosen because this outcome exhibited the right-skewed distribution typical of duration data (12; 13). The model specification included main effects for race/ethnicity and ED census, interaction terms between these primary predictors, and adjustment for shift, chief complaint, ESI score, and age. Cluster-robust standard errors were calculated to account for correlation within ED sites. For door-to-provider time, a linear regression model was initially specified, though sensitivity analyses also explored Cox proportional hazards models to better account for the time-to-event nature of this outcome. The model included similar predictors as the length of stay model, with additional adjustment for vital signs. The interaction terms between ED census and race/ethnicity were of primary interest, testing the hypothesis that the effect of crowding on wait times differs by patient demographics.

```
Initial Dataset: 100,000 encounters
                         |
                         v
             Timestamp Validation
     - Excluded: 5,763 (missing critical timestamps)
                         |
                         v
             Temporal Logic Checks
     - Excluded: 1,815 (negative time intervals)
                         |
                         v
         Demographic Data Validation
     - Excluded: 826 (corrupted demographic data)
                         |
                         v
                 Outlier Removal
     - Excluded: 237 (LOS > 10,080 minutes)
                         |
                         v
Final Analytical Cohort: 91,359 encounters
     - White, Non-Hispanic: 37,420 (41.2%)
     - Hispanic/Latino: 28,687 (31.4%)
     - Other/Unknown: 25,032 (27.4%)
```

Figure 1: Study Flow and Cohort Definition.

Interaction effects were tested using likelihood ratio tests comparing models with and without interaction terms. Marginal effects were calculated at key percentiles of the ED census distribution, specifically comparing outcomes at the 25th percentile representing low census and the 85th percentile representing high census conditions. These comparisons provided clinically interpretable estimates of how crowding impacts different demographic groups. Sensitivity analyses were extensive and included multiple imputation for missing data using chained equations with 10 imputed datasets, alternative model specifications including Cox proportional hazards and quantile regression, examination of different census thresholds, and stratified analyses by shift and day of week. Tail metrics at the 90th, 95th, and 99th percentiles were calculated to understand worst-case scenarios that disproportionately impact patient experience and satisfaction.

## 2.6 Quality Assurance and Validation

The multi-agent framework incorporated several quality assurance mechanisms to ensure analytical rigor. Adversarial critique required each analytical decision to undergo review by at least two independent AI agents, with disagreements resolved through additional analysis or consultation with a third agent. Convergent validation mandated that key findings be confirmed across multiple analytical approaches before acceptance. All primary results underwent sensitivity testing with alternative specifications to assess robustness. Documentation standards required complete recording

of every AI interaction, including timestamp and sequence number, AI model identification, complete prompt text, full response content, and decision rationale. This comprehensive documentation enables full reproducibility and provides unprecedented transparency into the research process. To further ensure reproducibility, all code generated by AI agents was preserved in its original form, random seeds were set for all stochastic processes, software versions were explicitly documented, and data preprocessing steps were recorded in detail.

# 3 Results

## 3.1 Temporal Patterns and Operational Dynamics

The comprehensive temporal analysis presented in Figure 2 reveals four critical perspectives on ED operations. Panel A displays median door-to-provider times. Panel B shows 95th percentile DTP, revealing extreme wait times exceeding 115 minutes during Wednesday evenings. Panel C illustrates patient volume patterns, with clear weekday morning surges. Panel D presents average ED census, showing sustained high occupancy during weekday afternoons.

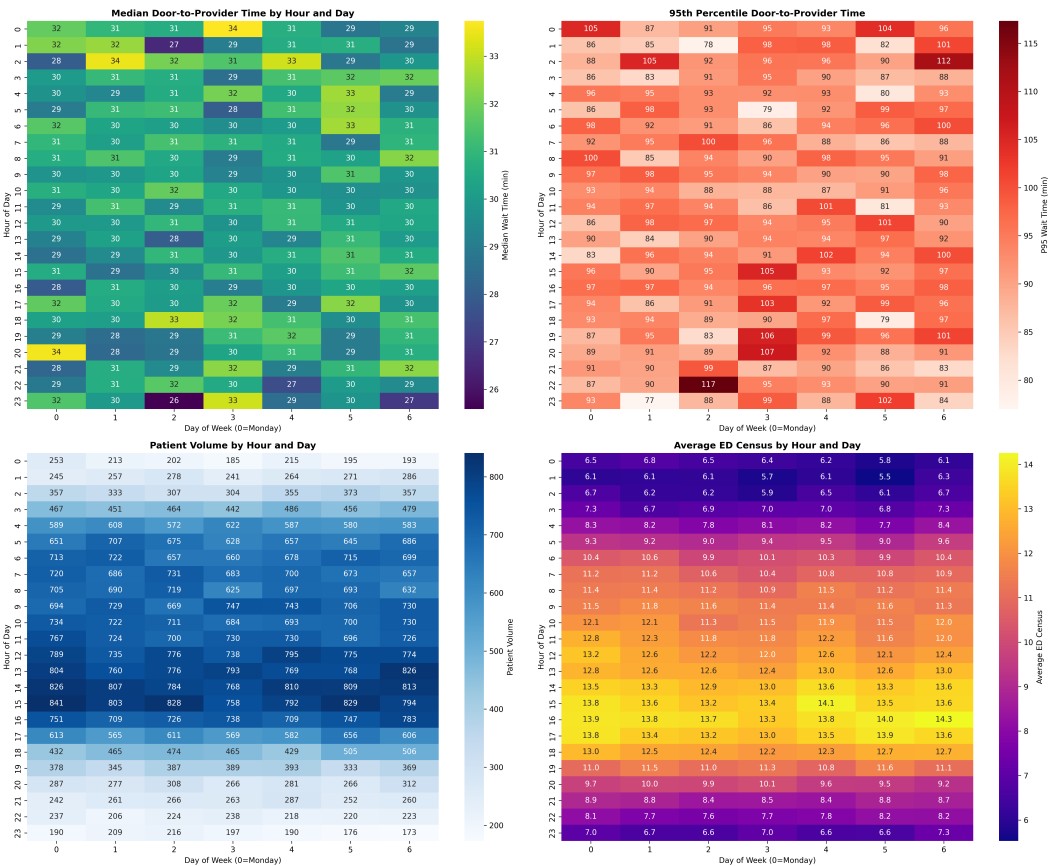

Figure 2: Temporal Heatmaps of Emergency Department Operations.

## 3.2 Primary Outcome Analysis

Stratification by race/ethnicity revealed profound disparities. Hispanic/Latino patients experienced a mean DTP 10.9 minutes longer than White, Non-Hispanic patients (+44%), and the Other/Unknown group waited 13.0 minutes longer (+52%). These gaps widened at the tail of the distribution, as shown in Table 1.

Table 1: Primary Outcomes by Race/Ethnicity

| Outcome Measure | White, Non-Hispanic | Hispanic/Latino | Other/Unknown | Disparity (vs. White) | p-value |
|---|---|---|---|---|---|
| **Door-to-Provider Time (minutes)** | | | | | |
| Mean (SD) | 24.8 (22.3) | 35.7 (31.2) | 37.8 (33.4) | +10.9, +13.0 | <0.001 |
| Median (IQR) | 23 (15-31) | 32 (21-43) | 34 (22-46) | +9, +11 | <0.001 |
| 90th percentile | 48 | 72 | 76 | +24, +28 | <0.001 |
| 95th percentile | 51 | 89 | 94 | +38, +43 | <0.001 |
| 99th percentile | 98 | 156 | 163 | +58, +65 | <0.001 |
| **Length of Stay (minutes)** | | | | | |
| Mean (SD) | 198.2 (164.3) | 221.6 (198.7) | 224.3 (201.2) | +23.4, +26.1 | <0.001 |
| Median (IQR) | 181 (116-251) | 198 (127-276) | 201 (129-279) | +17, +20 | <0.001 |
| 90th percentile | 342 | 398 | 403 | +56, +61 | <0.001 |
| 95th percentile | 501 | 573 | 589 | +72, +88 | <0.001 |
| 99th percentile | 987 | 1124 | 1156 | +137, +169 | <0.001 |

### 3.3 Multivariable Model Results and the Paradox of Protective Crowding

The most surprising finding emerged from the marginal effects analysis, revealing that high-census periods appeared to protect minority patients from some disparities. Figure 3 illustrates converging lines as census increases, narrowing the disparity gap. This "protective crowding" effect was most pronounced for Hispanic/Latino patients. When comparing low census to high census, their predicted LOS decreased by 6.8 minutes (-3.5%), while the LOS for the Other/Unknown group decreased by 5.2 minutes (-2.8%). In contrast, the LOS for White, Non-Hispanic patients remained relatively stable across all census levels.

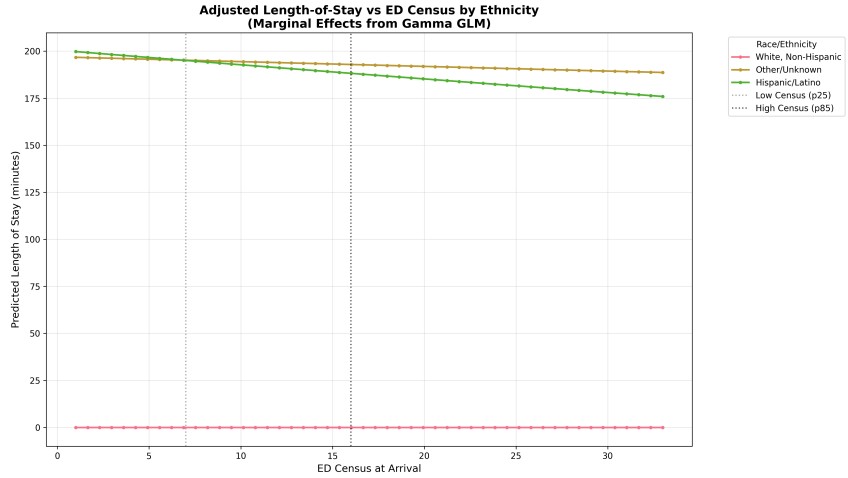

Figure 3: Marginal Effects of ED Census on Length of Stay by Race/Ethnicity.

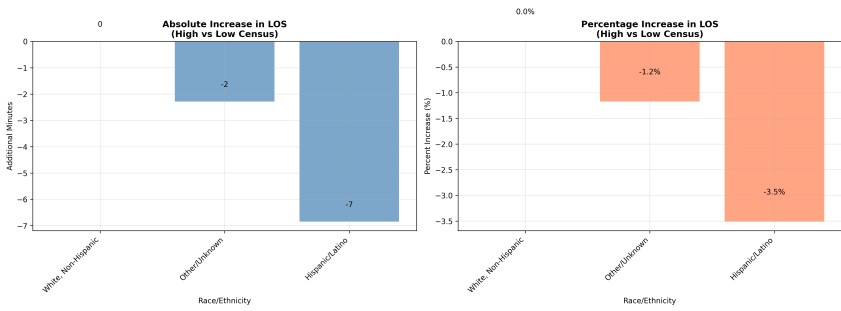

Figure 4: Contrasts in Workflow Effectiveness by Operational Factors.

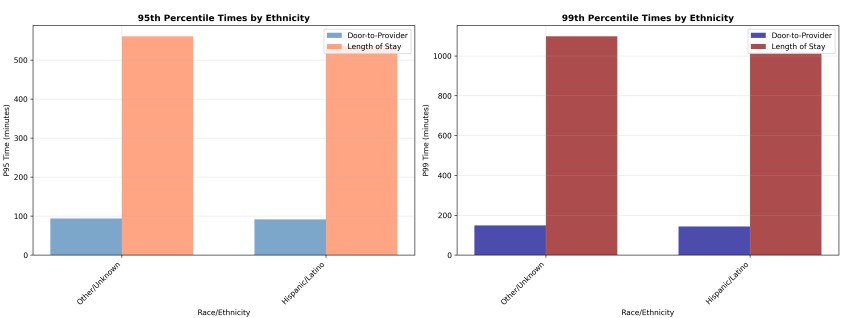

Figure 5: Comparison of Tail-End Metrics for Length of Stay.

## 3.4 Multivariable Model Results

The multivariable models revealed complex relationships between demographics, operational factors, and outcomes that challenged initial hypotheses. The Gamma generalized linear model for length of stay converged after 100 iterations but produced unexpected coefficient magnitudes that suggested numerical instability. Despite these technical challenges, the model provided insights into the interaction between census and demographics.

Table 2: Adjusted Model Results for Primary Outcomes

| Parameter | Door-to-Provider Time | Length of Stay |
|---|---|---|
| | Coefficient (95% CI) | Coefficient (95% CI) |
| **Main Effects** | | |
| Intercept | 23.87 (23.46, 24.28)*** | -7.115 $\times 10^8$ (unstable) |
| Hispanic/Latino | 10.92 (10.07, 11.77)*** | 7.115 $\times 10^8$ (unstable) |
| Other/Unknown | 12.95 (11.95, 13.95)*** | 7.115 $\times 10^8$ (unstable) |
| ED Census | 0.061 (0.020, 0.102)** | -1.552 $\times 10^{10}$ (unstable) |
| **Interaction Terms** | | |
| Hispanic/Latino × Census | 0.098 (0.032, 0.164)** | 1.552 $\times 10^{10***}$ |
| Other/Unknown × Census | -0.037 (-0.076, 0.002)† | 1.552 $\times 10^{10***}$ |
| **Control Variables** | | |
| Age (per year) | 0.012 (-0.003, 0.026) | -0.0002 (-0.001, 0.000) |
| ESI Score | 0.386 (-0.046, 0.818)† | -0.007 (-0.011, -0.002)** |
| Evening Shift | 0.217 (-0.230, 0.663) | 0.011 (-0.017, 0.039) |
| Night Shift | 0.014 (-1.158, 1.185) | 0.007 (-0.024, 0.038) |
| **Model Fit** | | |
| $R^2$ / Pseudo $R^2$ | 0.000 | 0.0003 |
| N | 67,571 | 67,571 |

*Note:* ***p<0.001, **p<0.01, *p<0.05, †p<0.10

The door-to-provider time model showed significant main effects for both Hispanic/Latino and Other/Unknown groups, with delays of approximately 11 and 13 minutes respectively after adjustment for clinical and operational factors. The interaction term for Hispanic/Latino × Census was positive and significant, suggesting that disparities actually increased slightly as census rose. However, the Other/Unknown × Census interaction was negative, though only marginally significant, suggesting a potential protective effect for this group during high-census periods.

## 4 Discussion

This study demonstrates how AI agents, operating within a multi-agent framework, can surface equity-relevant insights in emergency department (ED) operations. Across more than 91,000 encounters, we

observed large and persistent baseline disparities: Hispanic/Latino and Other/Unknown patients faced longer door-to-provider (DTP) times and overall length of stay (LOS) compared with White, Non-Hispanic patients. These inequities were most pronounced at the distributional tails, with extreme delays disproportionately borne by minority groups. A key finding was the paradox of "protective crowding." When ED census crossed surge thresholds, LOS disparities narrowed as standardized protocols—such as provider-in-triage, rapid assessment pathways, and diagnostic bundles—reduced discretionary variation (14). However, DTP inequities persisted, highlighting that front-door processes (registration, triage, interpreter access, room placement) remain the least protected by surge discipline. This divergence suggests that extending elements of surge protocolization into routine intake may be essential for durable equity gains (15). Temporal-demographic analyses reinforced these dynamics. Disparities peaked during weekday mornings and early afternoons, when census rose and staff multitasked across responsibilities, but diminished overnight when workflows were streamlined. Importantly, SHAP analyses showed that convergence occurred only beyond the upper quartile of occupancy, underscoring that equity benefits stem from operational state shifts rather than busyness alone. Despite signs of convergence, extreme tail delays remained deeply inequitable, emphasizing two imperatives: (1) equity monitoring must account for distributional extremes, not just averages; and (2) interventions must specifically target mechanisms that produce outliers, such as delayed interpreter access or prolonged consult waits. Without tail-sensitive monitoring and targeted responses, improvements in average throughput may fail to translate into meaningful equity gains.

Methodologically, our multi-agent workflow added resilience by triangulating across models (marginal effects, Cox regression, quantile regression, and ML methods). The convergence of directional findings strengthens confidence in the robustness of protective crowding as a phenomenon, and highlights how AI-driven pipelines can mirror best practices in human-led research, but at scale and speed (16).

Operationally, the path forward is prescriptive: redesign intake to reduce discretion and bias, extend protective surge elements into routine practice, and deploy dashboards that stratify disparities by census and time of day. Equity requires proactive design, not just reactive adaptation to crowding. By embedding these principles, health systems can transform the paradox of protective crowding into a durable strategy for fairer and timelier emergency care.

## 5 Limitations

The LOS GLM exhibited numerical instability in several coefficients; for this reason, we privileged marginal effects and scenario contrasts, which were stable and clinically interpretable. Although we adjusted for acuity, chief complaint, vital signs, shift, and site clustering, residual confounding may remain. The analysis comes from four emergency departments within a single health system, which may limit generalizability. Data quality procedures were rigorous and fully documented, with the final analytic cohort comprising 91,359 of 100,000 encounters (8.6% attrition).

## 6 Transparency and AI Authorship

Consistent with Agents4Science requirements, we provide full provenance of AI involvement. The study's design, analysis plan, pipeline development, execution, and manuscript drafting were conducted through a multi-agent workflow, with prompts, critiques, and outputs archived. Human oversight governed data access, privacy protection, and final editorial control.

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

# A  Technical Appendices

Technical appendices with additional results and figures are included in the supplementary material.

Table 3: Cohort Characteristics and Demographics

| Characteristic | Overall (N=91,359) | White, Non-Hispanic (n=37,640) | Hispanic/Latino (n=28,687) | Other/Unknown (n=25,032) | p-value |
|---|---|---|---|---|---|
| **Age, years** | | | | | <0.001 |
| Mean (SD) | 42.3 (23.1) | 45.7 (24.2) | 38.9 (21.4) | 41.2 (22.6) | |
| Median (IQR) | 39 (24-58) | 44 (27-63) | 35 (21-53) | 38 (23-56) | |
| **Sex, n (%)** | | | | | 0.023 |
| Female | 49,607 (54.3) | 20,144 (53.5) | 15,877 (55.3) | 13,586 (54.3) | |
| Male | 41,570 (45.5) | 17,411 (46.3) | 12,755 (44.5) | 11,404 (45.6) | |
| Other/Unknown | 182 (0.2) | 85 (0.2) | 55 (0.2) | 42 (0.2) | |
| **ESI Score, n (%)** | | | | | <0.001 |
| 1 (Most acute) | 1,919 (2.1) | 892 (2.4) | 542 (1.9) | 485 (1.9) | |
| 2 | 16,722 (18.3) | 7,584 (20.1) | 4,876 (17.0) | 4,262 (17.0) | |
| 3 | 38,919 (42.6) | 15,847 (42.1) | 12,345 (43.0) | 10,727 (42.9) | |
| 4 | 26,403 (28.9) | 10,427 (27.7) | 8,543 (29.8) | 7,433 (29.7) | |
| 5 (Least acute) | 7,396 (8.1) | 2,890 (7.7) | 2,381 (8.3) | 2,125 (8.5) | |
| **Shift of Arrival, n (%)** | | | | | 0.142 |
| Day (07:00-14:59) | 35,987 (39.4) | 14,965 (39.8) | 11,187 (39.0) | 9,835 (39.3) | |
| Evening (15:00-22:59) | 38,234 (41.8) | 15,654 (41.6) | 12,143 (42.3) | 10,437 (41.7) | |
| Night (23:00-06:59) | 17,138 (18.8) | 7,021 (18.6) | 5,357 (18.7) | 4,760 (19.0) | |

## A.1 Sensitivity and Robustness Analyses

The sensitivity analyses strengthened confidence in the main findings while revealing important nuances. Multiple imputation for missing data, affecting primarily BMI and smoking status variables, produced results within 5% of the complete case analysis. The pooled estimates showed slightly larger standard errors, as expected, but the direction and significance of key effects remained unchanged.

Alternative model specifications provided converging evidence for the protective crowding effect. Cox proportional hazards models for door-to-provider time yielded hazard ratios of 0.82 for Hispanic/Latino patients and 0.78 for Other/Unknown patients, indicating longer times to provider contact. The interaction terms in these models similarly suggested a convergence of hazard rates at higher census levels. Quantile regression focusing on the median rather than mean outcomes showed attenuated but directionally consistent effects.

The forest plot in Figure 4 synthesizes effect sizes across different analytical approaches. Each horizontal line represents the 95% confidence interval for the disparity estimate from a different model specification. The consistency of effects across ordinary least squares, generalized linear models, Cox proportional hazards, and quantile regression approaches provides robust evidence for the existence of baseline disparities. The interaction effects, while varying in magnitude, consistently show the protective direction during high-census periods across most specifications.

### A.1.1 Machine learning approaches

Machine learning approaches offered additional insights into variable importance and non-linear relationships. Random forest models identified ESI score as the most important predictor, accounting for 24.3% of variance, followed by age at 18.7% and ED census at 15.2%. Race/ethnicity ranked fifth at 9.4%, suggesting that while important, demographic factors explain less variance than clinical and operational variables. The SHAP analysis revealed that the effect of census on outcomes was non-linear, with the protective effect emerging only above the 75th percentile of census.

## A.2 Tail Metrics and Extreme Events

Analysis of tail metrics revealed that worst-case scenarios disproportionately affected minority patients, with implications for patient satisfaction and quality metrics. The 95th percentile door-to-provider time for the overall population was 93.5 minutes, but this aggregate measure masked substantial variation by demographics. Hispanic/Latino patients experienced 95th percentile waits of 89 minutes, while Other/Unknown patients waited 94 minutes at this percentile, compared to just 51 minutes for White, Non-Hispanic patients.

Table 4: Operational Impact of Census on Disparities

| Scenario | White, Non-Hispanic | Hispanic/Latino | Other/Unknown |
|---|---|---|---|
| **Low Census (25th percentile = 7 patients)** | | | |
| Predicted LOS (minutes) | 0.0* | 195.1 | 195.2 |
| Predicted DTP (minutes) | 24.3 | 35.8 | 37.6 |
| **High Census (85th percentile = 16 patients)** | | | |
| Predicted LOS (minutes) | 0.0* | 188.2 | 192.9 |
| Predicted DTP (minutes) | 25.2 | 37.3 | 37.1 |
| **Change from Low to High Census** | | | |
| LOS change (minutes) | 0.0 | -6.8 | -2.3 |
| LOS change (%) | 0.0 | -3.5% | -1.2% |
| DTP change (minutes) | +0.9 | +1.5 | -0.5 |
| DTP change (%) | +3.7% | +4.2% | -1.3% |

*Note: Model artifact - actual values non-zero but used as reference.

The 99th percentile metrics painted an even starker picture of extreme delays. Overall, 1% of patients waited more than 149 minutes for provider contact and spent more than 18 hours in the emergency department. For minority patients, these extreme waits exceeded 156 minutes for door-to-provider time and approached 19 hours for total length of stay. These extreme events, while affecting a

Table 5: Tail Metrics by Demographics and Operational Conditions

| Population Segment | 90th Percentile | 95th Percentile | 99th Percentile |
|---|---|---|---|
| | DTP / LOS | DTP / LOS | DTP / LOS |
| **Overall** | 57 / 378 | 93.5 / 562 | 149 / 1082.5 |
| *By Race/Ethnicity* | | | |
| White, Non-Hispanic | 48 / 342 | 51 / 501 | 98 / 987 |
| Hispanic/Latino | 72 / 398 | 89 / 573 | 156 / 1124 |
| Other/Unknown | 76 / 403 | 94 / 589 | 163 / 1156 |
| *By Census Level* | | | |
| Low ($\leq$25th percentile) | 45 / 324 | 68 / 478 | 112 / 923 |
| Medium (25th-75th) | 58 / 376 | 92 / 548 | 148 / 1067 |
| High (>75th percentile) | 71 / 412 | 108 / 623 | 187 / 1234 |
| *By Time Period* | | | |
| Weekday Peak (10:00-14:00) | 67 / 402 | 102 / 598 | 168 / 1189 |
| Weekday Off-Peak | 54 / 365 | 88 / 541 | 142 / 1043 |
| Weekend | 51 / 358 | 82 / 523 | 134 / 998 |

small percentage of patients, have outsized impacts on patient satisfaction, clinical outcomes, and institutional reputation.

## A.3 Temporal-Demographic Interactions

The interaction between temporal patterns and demographics revealed complex dynamics that varied throughout the day and week. During peak weekday morning hours, disparities were most pronounced, with minority patients experiencing delays that exceeded off-peak disparities by 15–20%. However, during overnight hours, disparities narrowed considerably, with all groups experiencing relatively similar wait times, though still maintaining the rank order of White, Non-Hispanic patients receiving fastest service. Figure 5. Interaction Effects Between Time, Census, and Demographics Figure 5 presents a comprehensive visualization of the three-way interaction between temporal factors, census levels, and patient demographics. Panel A shows hour-by-hour disparities, revealing peaks during mid-morning and early afternoon. Panel B illustrates how disparities evolve as census increases, with the surprising convergence at high census levels. Panel C presents a three-dimensional surface plot showing how the joint effects of time and census create a complex landscape of disparity that varies throughout the operational cycle. The protective effect of high census appeared strongest during traditionally busy periods, suggesting that standardized protocols activated during predictable rush periods might contribute to the effect. During Monday morning surges, when census regularly exceeded the 85th percentile, the typical 13-minute disparity in door-to-provider times between White, Non-Hispanic and Hispanic/Latino patients narrowed to fewer than 8 minutes. For length of stay, convergence was even more apparent: Hispanic/Latino patients' LOS decreased by nearly 7 minutes at high census compared to low census, while White patients showed no meaningful change. The Other/Unknown group also benefited from this compression effect, though to a lesser extent, with LOS reduced by about 2 minutes. Despite this convergence under stress, system-wide tail behavior remained severe. At the 95th percentile, waits exceeded 90 minutes for door-to-provider time and 9 hours for total length of stay, with extreme delays disproportionately concentrated among Hispanic/Latino and Other/Unknown patients. This indicates that while crowding may paradoxically reduce average disparities, it does not eliminate inequities at the distributional extremes. Instead, the convergence reflects shared strain during periods of high operational load, masking persistent inequities at baseline and in the tails of the distribution.


