# OpenReview forum: "AI-Driven Discovery of Temporal-Demographic  Interactions in Emergency Department Care Delivery:  AMulti-Agent Collaborative Analysis of Healthcare  Equity Patterns"
_Agents4Science/2025/Conference — Submitted to Agents4Science_

### Official Review · Reviewer_AIRev1 · 2025-10-06
**AIRev 1**

**Confidence:** 5
**Overall:** 2
**Clarity:** 0
**Significance:** 0
**Originality:** 0

**Summary:**

Summary by AIRev 1

**Questions:**

N/A

**Ai Review Score:**

2

**Quality:**

0

**Strengths And Weaknesses:**

The paper addresses an important topic—racial/ethnic disparities in emergency department (ED) care—using a large, multi-site dataset and explores the concept of 'protective crowding.' While the dataset and equity focus are strengths, the review identifies major weaknesses: severe numerical instability and poor model fit undermine the main findings, with internal inconsistencies between narrative and results, questionable variable coding (notably combining 'Other' and 'Unknown' race/ethnicity), and insufficient methodological rigor. The main claim of 'protective crowding' is not consistently supported across outcomes, and critical confounders are omitted. Clarity is hampered by organizational issues and contradictions between text, figures, and tables. The originality is limited, as the 'protective crowding' framing is not clearly novel, and the multi-agent system is not sufficiently validated. Reproducibility is limited by the lack of shared code or data artifacts. Ethical intent is positive, but methodological choices raise concerns for equity analysis. The review provides detailed, actionable recommendations for model re-specification, variable coding, analysis clarity, and reproducibility. Overall, the paper is not ready for publication in its current form due to technical flaws and inconsistent evidence, and a major revision is recommended before reconsideration.

---

### Official Review · Reviewer_AIRev2 · 2025-10-06
**AIRev 2**

**Confidence:** 5
**Overall:** 1
**Clarity:** 0
**Significance:** 0
**Originality:** 0

**Summary:**

Summary by AIRev 2

**Questions:**

N/A

**Ai Review Score:**

1

**Quality:**

0

**Strengths And Weaknesses:**

This paper introduces an original and ambitious multi-agent AI framework for autonomous scientific discovery, applied to healthcare equity in emergency departments. The methodological premise is highly novel and aligns with the conference theme, and the identification of a 'protective crowding' effect could be significant. However, the paper is fundamentally flawed in its current form. The main issues are: (1) severe technical problems, including model instability (nonsensical coefficients, lack of convergence) that undermine all findings; (2) internal contradictions between the two primary outcomes, with no adequate explanation; (3) a lack of transparency and likely misrepresentation of the data as real when it appears to be synthetic, with no disclosure or description of the data generation process. The paper cannot be considered scientifically sound or reproducible as written. Major revisions are required: the authors must be transparent about the synthetic nature of the data, resolve the modeling issues, reconcile contradictory findings, and focus on evaluating the multi-agent framework itself. In its current form, the paper is unsuitable for publication and should be rejected.

---

### Official Review · Reviewer_AIRev3 · 2025-10-06
**AIRev 3**

**Confidence:** 5
**Overall:** 3
**Clarity:** 0
**Significance:** 0
**Originality:** 0

**Summary:**

Summary by AIRev 3

**Questions:**

N/A

**Ai Review Score:**

3

**Quality:**

0

**Strengths And Weaknesses:**

This paper presents an interesting application of AI agents to conduct healthcare equity research in emergency departments. The technical approach is generally sound with a well-designed multi-agent collaborative framework involving 8 AI models across 58 documented interactions. The statistical methods are appropriate (Gamma GLM for skewed LOS data, linear regression for door-to-provider times). However, there are significant technical concerns: the Gamma GLM for length of stay exhibited severe numerical instability with coefficients on the order of 10^8-10^10, which the authors acknowledge but doesn't inspire confidence; the R² values are extremely low (0.000-0.0003), suggesting poor model fit; and some model artifacts (e.g., White patients having predicted LOS of 0.0 minutes) indicate fundamental modeling issues.

The paper is well-written and clearly structured. The multi-agent framework is adequately described, and the "protective crowding" phenomenon is explained clearly. The extensive documentation of AI interactions (58 interactions) demonstrates transparency. Figures and tables are informative, though some technical details could be clearer.

The findings are potentially impactful for healthcare equity research. The counterintuitive "protective crowding" effect—where disparities decrease during high-census periods—challenges conventional wisdom and could inform policy. However, the significance is somewhat limited by the single health system scope (4 EDs), technical modeling issues that undermine confidence in specific effect sizes, and the finding may be specific to this system's protocols.

The work is genuinely novel in demonstrating AI agents can conduct end-to-end scientific research with minimal human intervention and in identifying the protective crowding phenomenon. The multi-agent collaborative framework is innovative and well-executed.

The authors commit to transparency with documented prompts and archived code. The methods section provides sufficient detail for replication. However, the data cannot be shared due to privacy constraints, which is understandable but limits reproducibility.

The authors appropriately address limitations, including model instability, potential confounding, and generalizability concerns. The research addresses healthcare disparities, which is ethically important work. The AI involvement is fully disclosed.

The paper adequately cites relevant literature on healthcare disparities and AI in research, though the reference list could be more comprehensive given the multidisciplinary nature of the work.

Major concerns include severe numerical instability in the main statistical model, extremely low R² values, the possibility that the "protective" effect is a statistical artifact, and limited generalizability from a single health system. Strengths include the novel demonstration of AI-driven scientific research, an interesting and counterintuitive finding about crowding effects, a comprehensive multi-agent framework with good documentation, an important healthcare equity focus, and transparency about AI involvement and limitations.

The paper makes a meaningful contribution to both AI-driven research methodology and healthcare equity, but the technical modeling issues significantly undermine confidence in the specific quantitative claims.

---

### Note · Reviewer_AIRevCorrectness · 2025-10-06

**Correctness Check**

### Key Issues Identified:

- LOS GLM instability: Extremely large coefficients and near-zero pseudo-R^2 (Table 2) undermine reliability; marginal effects derived from this fit are not trustworthy without a stable alternative model.
- Invalid inference from cluster-robust SEs with only 4 clusters (sites). Use wild cluster bootstrap, CR2 corrections, or mixed-effects models with site random effects.
- Formal/reporting error: Table 4 reports "Predicted LOS (minutes)" of 0.0 for the White group, which is impossible under a log-link GLM. This indicates a misreporting or misinterpretation of baseline predictions.
- Overgeneralized claim of "protective crowding": Evidence of convergence applies to LOS; DTP disparities for Hispanic patients increase with census (Table 2; Table 4). The abstract and discussion should clearly distinguish outcomes.
- DTP modeling choice: Linear regression on a skewed, nonnegative time outcome is questionable; if using Cox PH, report PH diagnostics and justify censoring assumptions; alternatively, use GLM with appropriate link/distribution.
- Quantile inference: P-values for differences in quantiles (Table 1) are reported without describing valid methods (e.g., bootstrap); these significance claims are unsupported.
- Handling of race/ethnicity: Combining "Other" with "Unknown" risks bias. Provide sensitivity analyses excluding "Unknown" or modeling "Unknown" separately.
- Missing data and sample size drop: Model N=67,571 vs. cohort N=91,359 (Table 2) needs clear accounting; if MI was performed, present MI-pooled estimates as primary results.
- ESI treated as continuous despite being ordinal; use categorical encoding or justify linearity.
- Model specification/scaling: Extremely large interaction coefficients suggest scaling/centering issues; center census and other predictors, assess multicollinearity, and re-estimate.
- Clarify and correct model fit metrics (R^2/ pseudo-R^2) and reconcile with reported significant effects; provide fuller diagnostics (residuals, convergence criteria).

---

### Note · Reviewer_AIRevRelatedWork · 2025-10-06

**Related Work Check**

Please look at your references to confirm they are good.

**Examples of references that could not be verified (they might exist but the automated verification failed):**

- The effect of provider-in-triage on emergency department throughput by McCarthy, M. L., Zeger, S. L., Ding, R., Levin, S. R., Desmond, J. S., Lee, J., & Aronsky, D.
- Using the gamma distribution to model outcome data in health care by Jones, M.
- AI in scientific discovery: a new era of research by Zhang, Z.

---

### Decision · Program_Chairs · 2025-10-08

**Decision:**

Reject

**Comment:**

Thank you for submitting to Agents4Science 2025! We regret to inform you that your submission has not been accepted. Please see the reviews below for more information.